# Mineral Metabolism in Children: Interrelation between Vitamin D and FGF23

**DOI:** 10.3390/ijms24076661

**Published:** 2023-04-03

**Authors:** Oscar D. Pons-Belda, Mª Agustina Alonso-Álvarez, Juan David González-Rodríguez, Laura Mantecón-Fernández, Fernando Santos-Rodríguez

**Affiliations:** 1Hospital Can Misses, 07800 Ibiza, Spain; oscardavid.pons@asef.es; 2Hospital Universitario Central de Asturias, 33011 Oviedo, Spain; maruchi.al.al@gmail.com (M.A.A.-Á.); laura_mantecon@hotmail.com (L.M.-F.); 3Hospital General Universitario Santa Lucía, 30202 Cartagena, Spain; juandavid.gonzalez@um.es; 4Department of Medicine, Faculty of Medicine, University of Oviedo, 33003 Oviedo, Spain

**Keywords:** fibroblast growth factor 23 (FGF23), vitamin D, children, rickets

## Abstract

Fibroblast growth factor 23 (FGF23) was identified at the turn of the century as the long-sought circulating phosphatonin in human pathology. Since then, several clinical and experimental studies have investigated the metabolism of FGF23 and revealed its relevant pathogenic role in various diseases. Most of these studies have been performed in adult individuals. However, the mineral metabolism of the child is, to a large extent, different from that of the adult because, in addition to bone remodeling, the child undergoes a specific process of endochondral ossification responsible for adequate mineralization of long bones’ metaphysis and growth in height. Vitamin D metabolism is known to be deeply involved in these processes. FGF23 might have an influence on bones’ growth as well as on the high and age-dependent serum phosphate concentrations found in infancy and childhood. However, the interaction between FGF23 and vitamin D in children is largely unknown. Thus, this review focuses on the following aspects of FGF23 metabolism in the pediatric age: circulating concentrations’ reference values, as well as those of other major variables involved in mineral homeostasis, and the relationship with vitamin D metabolism in the neonatal period, in vitamin D deficiency, in chronic kidney disease (CKD) and in hypophosphatemic disorders.

## 1. Major Regulatory Hormones of Mineral Metabolism

The mineral metabolism is essential for the homeostasis of calcium and phosphorus and to maintain healthy bones. The main regulatory hormones of the mineral metabolism are the parathyroid hormone (PTH), 1,25-dihydroxyvitamin D (1,25(OH)_2_D) and fibroblast growth factor 23 (FGF23), which interact in a complex, multi-tissue feedback system, graphically represented in Figure 1 [1].

PTH is synthesized by the parathyroid glands as a pre-pro-PTH of 115 amino acids. After intracellular processing, intact PTH (84 amino acids) is secreted, stored, or degraded intracellularly. Circulating calcium is the primary acute physiologic regulator of PTH secretion, which is stimulated by a decrease of calcemia, low 1,25(OH)_2_D levels and phosphate retention, whereas FGF23 has been shown to inhibit PTH secretion in animal and in vitro experiments [1].

In humans, vitamin D is mostly endogenously produced through the exposure of skin to sunlight whereas 10% or less, unless oral supplements are used, is absorbed in the gastrointestinal tract from a few natural foods such as fatty fish, cod liver oil, egg yolks, or some mushrooms and yeasts. Cholecalciferol produced in the skin is metabolized to 25-hydroxyvitamin D (25OHD) in the liver by a cytochrome P450 enzyme or vitamin D 25-hydroxylase. The biologically active hormone is 1,25(OH)_2_D or calcitriol, which is synthesized in the kidney from 25OHD by a cytochrome P450 enzyme or 1α-hydroxylase. Circulating concentrations of 1,25(OH)_2_D are tightly regulated by PTH, phosphate, calcium, FGF23 and 1,25(OH)_2_D itself [1,2]. There may be extrarenal activity of 1α-hydroxylase and the majority of cell types in the body express the vitamin D receptor (VDR). Thus, it has been known since the 1980s that the functions of vitamin D are not only restricted to the mineral metabolism and bones, but vitamin D also exerts important, although ill-defined, pleiotropic effects on numerous cell functions such as proliferation, differentiation, apoptosis, migration and reparative mechanisms [3].

FGF23 was first identified in the year 2000 by Yamashita et al. [4] in the ventrolateral thalamic nucleus of the brain. Further studies found that osteocytes and osteoblasts of bone are the main source of FGF23, although a small production has also been detected in salivary glands, stomach, skeletal muscle, mammary gland, liver, kidney and heart [5]. The pro-FGF23 protein is composed of three signal, N-terminal and C-terminal peptides of 24, 154 and 71 amino acids, respectively. After the cleavage of the signal peptide, O-glycosylation occurs through the enzyme GALTN3, which protects the protein from proteolysis and facilitates its incorporation into the bloodstream. The C-terminal fragment arises from proteolytic cleavage by furins that recognize a consensus site that inactivates and degrades active FGF23 protein. Only the intact form of FGF23 (iFGF23) is active. The C-terminal fragment (cFGF23) is essential for the interaction with the α-Klotho coreceptor, which increases the affinity of FGF23 for its receptor, confers stability to the union and favors signaling [6]. The main regulators of FGF23 secretion are phosphate, PTH, 1,25(OH)_2_D and calcium [1,7].

α-Klotho is a protein that consists of two extracellular domains (KL1 and KL2), a transmembrane domain and a smaller cytosolic domain. FGF23 binds to α-klotho through specific sites in the KL1 and KL2 domains of FGF23-expressing tissues such as the kidney, parathyroid gland and cerebral choroid plexus [8].

## 2. Circulating Variables Linked to Mineral Metabolism_Reference Values in the Pediatric Age

Clinical interpretation of biological variables’ laboratory measurements requires the availability of reliable reference intervals. According to the Clinical and Laboratory Standards Institute (CLSI) guidelines [9], a reference interval is the range that encompasses the central 95% of the distribution of a test and the results from individuals sampled from a healthy reference population. Comparing a given result with its appropriate reference interval is necessary for proper clinical interpretation. The process of establishing accurate and reliable reference intervals is complex and highly dependent on the selection of an appropriate reference population [10]. Factors such as age, sexual development, geographic location, sex and ethnicity may affect the reference concentration of a given analyte. Therefore, group-separated reference intervals that consider the influence of these covariates are needed for some analytes. This is particularly important for pediatric populations, as the concentrations of some routinely measured analytes vary significantly with growth and development [11,12]. Thus, the use of inappropriate pediatric reference intervals can result in misdiagnosis and/or misclassification of disease.

There are limited data about mineral metabolism variables in healthy children due to the complexity of the sample that involves changes inherent to growth and development and the difficulties involved in studies in the pediatric population compared to the adult population (difficulties in recruiting a large number of healthy participants and issues in collecting adequate blood volumes, particularly from very young children). Table 1 provides information on calcium, phosphate, PTH, vitamin D metabolites, FGF23 and Klotho in healthy children on the basis of published findings meeting the necessary criteria to be considered as reference laboratory values [13,14,15,16,17,18,19,20,21,22].

## 3. FGF23 and Vitamin D Metabolism

FGF23 and vitamin D reciprocally interact in a way that is not clear in both physiological conditions and several diseases. As graphically shown in Figure 1, 1,25(OH)_2_D directly stimulates the secretion of FGF23 by the osteocytes. Thus, FGF23 increases phosphaturia, counteracting the hyperphosphatemic effect resulting from the increased intestinal absorption of phosphorus caused by vitamin D. In addition, FGF23 decreases the renal synthesis of 1,25(OH)_2_D by inhibiting the 1 ɑ-hydroxylase activity, that transforms 25OHD in 1,25(OH)_2_D, and by stimulating the enzyme 24-hydroxylase, that degrades 1,25(OH)_2_D. 24-hydroxylation of both 25OHD and 1,25(OH)_2_D are processes of vitamin D inactivation and degradation. However, 24,25-dihydroxyvitamin D (24,25(OH)_2_D) is attributed a specific role in normal bone integrity and the healing of fractures [23]. FGF23 is considered the main counterregulatory hormone of vitamin D, but the relevance of these interactions in a healthy child is not well defined.

### 3.1. FGF23 and Vitamin D Metabolism: Fetal Life and Neonatal Period

Fetal development is characterized by rapid bone growth and mineralization. The placenta actively transports calcium, phosphorus and magnesium to maintain high circulating concentrations of these elements and facilitate bone mineralization [24]. FGF23 is produced by osteoblasts and osteocytes in fetal bone but does not contribute importantly to fetal regulation of serum phosphorus, placental phosphorus transport or skeletal development and mineralization. FGF23 produced by the fetus does not regulate placental phosphorus transport, despite the expression of FGF23 target genes in the placenta, nor does maternal FGF23 cross the placenta [25]. Moreover, neither 1,25(OH)_2_D nor PTH contribute to maintaining serum phosphorus levels [25]. In animal models, it has been demonstrated that the absence of either FGF23 or its coreceptor Klotho does not significantly affect fetal phosphorus metabolism and that the excess of FGF23 causes a moderate reduction in serum 1,25(OH)_2_D [26].

After birth, iFGF23 increases fourfold within the first 4–5 days of age whereas cFGF23 maintains the high values of fetal life [27,28], which suggests that FGF23 is produced but rapidly fragmented in the uterus, resulting in elevated levels of nonfunctional cFGF23 fragment. Cord blood levels of Klotho are sixfold adult and neonatal values [28]. It may contribute to reducing circulating iFGF23 levels [29].

Excess of FGF23 exerts its effects as early as 12 hours after birth, whereas the loss of Klotho or FGF23 does not have a noticeable biochemical effect until 5 to 7 days after birth [26]. The deficiency or excess of FGF23 usually does not cause clinical disease in the first months of life, although clinical manifestations have been reported in a few human case reports and in animal models (Table 2) [26,30,31,32,33,34]. Thus, it is unclear when FGF23 starts to play a significant role in the regulation of postnatal phosphorus metabolism.

Three studies providing reference mean (SD) values of FGF23 in the neonatal period are available. Ali et al. [27] find that plasma concentrations in 64 healthy term neonates are 16.7 (24.2) pg/mL for iFGF23 and 1678 (1857) RU/mL for cFGF23. Holmlund-Suila et al. [35] report iFGF23 and cFGF23 values in the blood cord at the birth of 113 healthy newborns: 3.0 (10.7) pg/mL and 536.2 (731.7) RU/mL in females and 3.0 (2.3) pg/mL and 605.9 (842.9) RU/mL in males. Takaiwa et al. [28] show an elevation of iFGF23 from low levels in cord blood to reach adult values on day 5 after birth. They measure FGF23 in 22 healthy term infants in cord blood or in 22 healthy term infants at 5 days of life and compare these ranges with healthy adults. iFGF23 levels are 3.9 (1.6) and 21.8 (17.6) pg/mL at birth and at 5 days of life, respectively, and cFGF23 concentrations are 73.3 (22.4) and 81.0 (28.2) RU/mL at birth and at 5 days of life, respectively. To our knowledge, there are no published studies in the neonatal period that correlate FGF23 levels to vitamin D or any of its metabolites.

### 3.2. FGF23 and Vitamin D Metabolism: Vitamin D Deficiency_Rickets

Factors that hinder the cutaneous synthesis of vitamin D, such as insufficient exposure to sunlight because of indoor confinement or regular use of clothes that cover the body (burka, chador, niqab), high latitude and dark skin all prevent adequate insolation and facilitate the occurrence of vitamin D deficiency. Malabsorptive gastrointestinal disorders, liver disease or the use of certain drugs able to decrease synthesis of vitamin D or increase its degradation also lead to a predisposition to vitamin D deficiency [36]. Insufficient intake or intestinal absorption of calcium is frequently associated with vitamin D deficiency, especially in malnourished populations [37]. 

Biochemically, vitamin D deficiency is identified by low serum concentrations of 25OHD. Although the measurement of free bioavailable fractions of 25OHD in serum might have some clinical interest in selected populations [38], total circulating 25OHD concentrations are considered the best marker of vitamin D status [39,40]. In 2011, the clinical practice guidelines of the Endocrine Society Task Force on Vitamin D [41] defined a cutoff serum 25OHD value of 50 nmol/L (2.5 nmol/L = 1 ng/mL) as indicative of vitamin D deficiency. The “Global Consensus Recommendations on Prevention and Management of Nutritional Rickets’’ recommends the following classification of vitamin D status based on serum 25OHD values: sufficiency, >50 nmol/L or >20 ng/mL; insufficiency, 30–50 nmol/L or 12–20 ng/mL; deficiency, <30 nmol/L or <12 ng/mL [42]. 

The classical, well-known major manifestation of vitamin D deficiency in children is rickets, which is defined as the lack of growth-plate mineralization, resulting from abnormal differentiation and maturation of chondrocytes and a marked disruption of chondrocyte columns. Defective mineralization of the synthesized osteoid when the growth plates have fused is osteomalacia, the term used in adults.

The clinical presentation of rickets includes skeletal (Table 3) and non-skeletal manifestations such as anemia, recurrent infections, decreased muscle tone, dental alterations (delayed eruption, caries, enamel alterations), stunted growth, carpopedal spasms and laryngospasm [43,44]. The irritability caused by bone pain may also be a symptom in infants, as well as, in severe cases, hypocalcemic complications such as tetany, seizures, dilated cardiomyopathy, and even death [45,46]. Most children with rickets caused by vitamin D deficiency have serum 25OHD concentrations less than 10 ng/mL and usually less than 5 ng/mL [47]. Serum calcium and phosphate levels may be normal or low whereas alkaline phosphatase and PTH are elevated. Vitamin D doses for treatment of rickets are shown in Table 4 [42]. Oral calcium administration, 500 mg/day, in conjunction with vitamin D is usually recommended regardless of age or weight. However, a recent systematic review did not find published evidence to support or discourage the addition of calcium to vitamin D for the treatment of nutritional rickets [48]. Intravenous calcium therapy is reserved for acute symptomatic hypocalcemia.

Despite the metabolic connection between FGF23 and vitamin D, little is known about the status of FGF23 and Klotho in individuals with rickets induced by vitamin D deficiency. A study in 492 healthy Japanese adolescents found that 28% are vitamin D deficient and there is no significant correlation between serum concentrations of 25OHD and FGF23 [21]. In contrast, Kubota et al. [49] found low serum FGF23, less than 19 pg/mL, in patients with vitamin-D-deficient rickets. Serum FGF23 and phosphate levels increase following vitamin D treatment. They suggested that low serum FGF23 might serve as a biochemical marker of vitamin-D-deficient rickets.

### 3.3. FGF23 and Vitamin D Metabolism: Vitamin D Deficiency_Extraskeletal Manifestations

In the last decades, there has been a growing interest in the role of vitamin D deficiency in many systemic diseases. Numerous epidemiological studies have found a high prevalence of low serum 25OHD concentrations in patients, mostly adults, with cancer, asthma, mental disorders, autoimmune disorders, diabetes, cardiovascular diseases, infections including COVID-19, etc. [50]. However, rigorous evidence showing a beneficial effect of vitamin D supplementation on the outcome of these diseases in randomized controlled trials is lacking [51]. The most significant studies are summarized in Table 5 [52,53,54,55,56,57,58,59,60,61,62,63,64,65,66].

Inflammation is a common feature of chronic diseases and has a major impact on morbidity and mortality. Vitamin D and FGF23/Klotho interact reciprocally and may have significant effects on immunity. As stated above, vitamin D deficiency is highly prevalent in systemic diseases and may lead to the loss of vitamin-D-dependent anti-inflammatory actions, while concentrations of FGF23 are characteristically high in chronic disorders such as inflammatory bowel disease or CKD (see below). FGF23 may affect indirectly immunity through inhibition of vitamin D production, and by direct effects on myeloid cells impairing immune cell functions [67]. The factor kappa-light-chain-enhancer of activated B cells (NF-kB), implicated in multiple inflammatory processes, may in turn activate the transcription of FGF23 [68,69]. Klotho has been shown to act as a tumor suppressor, due to its down-regulatory effect on major prosurvival signaling cascades required for cancer progression. Some studies have also shown a positive relationship between FGF23 concentration and insulin resistance [70,71] and an association of FGF23 circulating levels with endothelial dysfunction, vascular calcification, left ventricular hypertrophy and incidence of mortality and cardiovascular events in the general population [72], although low levels of 25OHD are usually also associated with these occurrences [73]. As for the effect of vitamin D supplementation on serum FGF23 concentration in individuals with vitamin D deficiency, many questions remain unanswered [74,75]. Trummer et al. [76] report that vitamin D supplementation has no significant effect on FGF23 in a randomized controlled trial conducted in subjects with arterial hypertension, however, they do observe an increase of FGF23 concentrations in individuals with serum 25OHD below 20 ng/mL. In contrast, Bhagatwala et al. [77] fail to demonstrate this effect in a similar study. A systematic review and meta-analysis published in 2019 by Charoenngam et al. [75] concludes that vitamin D3 supplementation leads to a significant increase in serum-intact FGF23 in patients with vitamin D deficiency diagnosed by serum 25OHD < 20 ng/mL.

### 3.4. FGF23 and Vitamin D Metabolism: Chronic Kidney Disease

CKD invariably affects the mineral metabolism, causing distinctive alterations in pediatric patients because the array of diseases leading to chronically reduced glomerular filtration rate (GFR) is different from adults, the tubulointerstitial and congenital nephropathies being predominant, and because there are marked differences in the osseous metabolism between children and adults, growth and endochondral ossification being exclusive to the pediatric age [78,79]. 

The term bone mineral disorder in CKD (CKD-MBD) describes one or a combination of the following three components: (i) Abnormalities of calcium, phosphorus, PTH, FGF23 and vitamin D metabolisms; (ii) Bone abnormalities (short stature, reduced mineralization and increased risk of fractures); (iii) Extraskeletal calcification [80]. CKD-MBD is a systemic disorder that involves not only regulatory hormones of calcium and phosphate homeostasis but also keeps an important and not well-understood connection with endocrine and paracrine factors regulating body nutrition and longitudinal growth [81]. In addition, CKD-MBD is related to the risk of cardiovascular complications in advanced CKD, the most prevalent cause of mortality in these patients. An adequate prevention and treatment of CKD-MBD in childhood is expected to improve the long-term outcome and cardiovascular morbidity.

Several studies demonstrate high circulating levels of FGF23 in adult [82] and pediatric [83,84] patients with CKD and reduced GFR. Some characteristics of elevated serum FGF23 in children and adolescents with CKD are schematically shown in Table 6 [83,84,85,86,87,88,89,90,91,92,93,94]. The interplay between FGF23 and vitamin D and other mineral metabolisms regulating hormones in CKD is not well understood. Although clinical studies do not show coincident findings, it may be concluded that serum FGF23 correlates directly with phosphorus, PTH and calcium—phosphorus product correlates inversely with 1,25(OH)_2_D, and shows no consistent correlation with calcium or 25OHD (Table 6). Whereas 1,25(OH)_2_D appears to be a potent stimulator of FGF-23 synthesis, FGF23 suppresses 1,25(OH)_2_D kidney production and increases the catabolism of both 1,25(OH)_2_D and 25OHD [89,95]. Few data have been published on the effect of treatment with vitamin D metabolites on FGF23 concentrations. In a multicenter European trial, administration of ergocalciferol for 8 months further increases serum FGF23 values in vitamin-D-deficient children with advanced CKD [96]. Intermittent oral cholecalciferol supplementation increases serum 25OHD but does not modify PTH or FGF23 levels in French children and teenagers with CKD, kidney transplantation or stable nephrotic syndrome [97]. In a cohort of Indian patients, 600,000 IU of cholecalciferol over 3 days increases serum FGF23 and phosphate levels in CKD stage 2 but not in CKD stages 3 and 4 [98].

High serum FGF23 concentrations likely result from an increased production of FGF23 in osteocytes and from decreased renal clearance [99,100]. Although the exact contribution of each of these two mechanisms is unknown, anuric subjects have been shown to have higher FGF23 values than patients with any degree of residual renal function. Proteinuria induces elevation of both plasma phosphate and FGF23 concentrations independent of GFR [101] and it is controversial whether, for a similar degree of GFR reduction, patients with glomerular diseases have higher serum levels of FGF23 than those with CKD caused by other nephropathies [89,90]. Corticosteroids increase serum FGF23, likely by a direct stimulating effect of its synthesis by osteocytes [84], but no differences in circulating FGF23 values have been found in transplant recipients as compared to GFR-matched patients with CKD in their native kidneys [102]. 

The clinical consequences of the increased FGF23 levels still need to be clarified. Given the stimulating direct effects of FGF23 on the development of left ventricular hypertrophy via the FGF receptor 4, and on sodium retention in the kidney tubule [103,104], the excess of FGF23 has been proposed as a pathogenic factor of the high cardiovascular morbidity and mortality found in CKD patients [85,105,106]. However, clinical studies and meta-analysis of publications indicate that the link between high FGF23 and cardiovascular complications is a noncausal association between two prevalent findings in CKD rather than a cause—effect relationship [107,108,109]. 

The elevated FGF23 concentrations may represent an independent risk factor for the progression of CKD. Portale et al. [93] find that the period of time to start dialysis or kidney transplantation or 50% decline from baseline GFR is 40% shorter for CKD children in the highest compared with the lowest FGF23 tertile. Likewise, higher FGF23 levels are independently associated with biopsy-proven chronic renal allograft injury in a cross-sectional, multicenter, case-control study on transplanted children [110] and FGF23 values predict future episodes of rejection in pediatric renal allograft recipients [111]. Thus, the adverse effect of FGF23 on kidney function, GFR deterioration rate and the risk of graft rejection in CKD children needs to be confirmed as well as clarifying the potential underlying mechanisms. 

The effects of high FGF23 on bone metabolism and longitudinal growth are not well understood despite being of high interest in pediatric CKD. High circulating levels of FGF23 are associated with improved indices of skeletal mineralization analyzed by bone biopsy in pediatric patients on chronic peritoneal dialysis [83] but not in another group of pediatric patients with predialysis CKD [86]. Interestingly, in this last study, an association exists between FGF23 and height Z score and Bacchetta et al. [112] also find a positive correlation between FGF23 and IGF1 levels in children with normal renal function. However, an inverse relationship between FGF23 levels and height Z score has been reported in children on chronic peritoneal dialysis [92]. Thus, there are no uniform clinical findings on a potential relationship between FGF23 and growth, which is not surprising because endochondral growth is complex and depends on many local and systemic factors whose regulation is impaired in CKD. Animal models of hypophosphatemia indicate that the excess of FGF23 exerts a marked adverse effect on growth plate structure and dynamics and that blocking FGF23 action reverses this action and stimulates longitudinal growth as well as formation and mineralization of bone [113]. In CKD, studies investigating the role of FGF23 on endochondral growth are not available.

### 3.5. FGF23 and Vitamin D Metabolism: Hypophosphatemic Disorders

Rickets is a heterogeneous group of diseases resulting in disturbances in calcium and/or phosphate homeostasis, thereby affecting the growing skeleton. Clinical manifestations depend on the age at the onset and the duration of the disease (Table 3). Additional features may be present depending on the underlying disease, e.g., craniosynostosis, dental abscesses, hearing loss and growth delay in X-linked hypophosphatemia (XLH) [114,115]. 

Historically, rickets was classified as calcipenic or phosphopenic rickets. However, growing scientific evidence suggests that the ultimate cause of rickets is due to insufficient availability of phosphate necessary for the mineralization and terminal differentiation of growth plate chondrocytes [116]. Serum phosphate levels are mainly regulated by PTH, 1,25(OH)_2_D and FGF23, although other phosphatonins are known (MEPE, sFRP4 and FGF7). [117] The knowledge of its underlying mechanisms has been substantially improved by the description of ‘novel’ genetic diseases over the last decades. Currently, many causes of acquired and inherited rickets have been identified which show a considerable overlap in their clinical findings. (Table 7) [114,115,116].

In patients with untreated calcemic rickets, markedly elevated PTH levels may be observed in order to maintain normal serum calcium levels. In contrast, in patients with phosphopenic rickets, PTH levels may be observed in the normal range, but in patients with FGF23-driven phosphopenic rickets, PTH levels may be slightly elevated because FGF23 suppresses 1,25(OH)_2_D levels, which in turn stimulates PTH secretion by the parathyroid glands. In addition, low vitamin D levels are frequently observed in the general population. Otherwise, PTH levels are often suppressed in patients with SLC34A1 and SLC34A3 mutations, due to 1,25(OH)_2_D excess [114,115,118]. In patients with FGF23-mediated rickets, 1,25(OH)_2_D levels are usually low or inappropriately normal in the setting of hypophosphatemia. [114,115,118].

The measurement of FGF23 levels is most useful in the diagnostic workup of untreated patients with phosphopenic rickets to differentiate between FGF23-mediated and other forms of rickets. FGF23 levels are elevated or inappropriately normal in patients with several inherited hypophosphatemic disorders and tumor-induced osteomalacia (TIO) which normalized after tumor resection [119,120]. The combination of hypophosphatemia in children and adults and intact FGF23 concentrations greater than 30 pg/mL allows for the identification of patients with FGF23-mediated disorders [121]. Another recent study shows similar results in adults, with an intact FGF23 cut point of 27 pg/mL distinguishing FGF23-mediated from FGF23-independent hypophosphatemia, and a cFGF23 cut point of 90 RU/mL identifying specifically TIO from FGF23-independent hypophosphatemia [122].

It should be noted that the kidney is central to the regulation of phosphate homeostasis. Renal tubular reabsorption of phosphate in the proximal tubule is mainly mediated by sodium-phosphate cotransporters (NaPi-2a, NaPi-2c and PiT-2). PTH and FGF23 increase phosphaturia at the proximal tubule level by reducing the expression of NaPi-2a and NaPi-2c cotransporters. In addition, it has been suggested that vitamin D action on renal Pi handling could be indirect, involving altered serum levels of FGF23 [115,123,124].

FGF23 participates in phosphate homeostasis through the binding of the FGF receptor 1 and coreceptor α-Klotho, which is highly expressed in the kidney and parathyroid gland. The soluble form of Klotho can also act as a coreceptor for FGF23 signaling [117]. In addition, Klotho can modulate PTH secretion and decrease the abundance of NaPi-2a in the proximal tubule and thus may act as a phosphaturic factor. FGF23 and Klotho are also able to increase tubular calcium reabsorption through TRPV5. In addition, as previously mentioned, FGF23 decreases the renal synthesis of 1,25(OH)_2_D and regulates PTH synthesis and secretion through MAPK pathways. In another way, FGF23 overexpression in vitro can suppress not only osteoblast differentiation but also matrix mineralization, regardless of its systemic effect on phosphate metabolism [124,125,126].

Although the role of FGF23 in normal physiology is not clear, FGF23 synthesis is shown to be stimulated by oral phosphorus loading and inhibited by phosphate restriction in healthy subjects [127]. In addition, the discovery of the important role of FGF23 in the bone-kidney-parathyroid axis has led to a better understanding of the genetic conditions associated with phosphate disorders, the most common cause of phosphopenic rickets being XLH. 

XLH is caused by loss-of-function mutations in the X-linked gene for phosphate regulatory endopeptidase (PHEX) and although its transmission follows dominant X inheritance, many cases are caused by de novo mutations. These inactivating mutations of PHEX cause an increased concentration of FGF23, leading to hyperphosphaturia, hypophosphatemia and low or inappropriately normal serum 1,25(OH)_2_D concentrations [128]. Table 8 summarizes recent and significant studies published in humans with hypophosphatemic disorders about the effects of treatment and the possible interaction between FGF23, Klotho and vitamin D [129,130,131,132,133,134,135,136,137,138,139,140,141,142,143,144,145,146,147,148,149,150,151,152,153,154,155,156,157,158,159,160]. Several mechanisms are still currently unclear and further studies are needed to explore the selective targeting of the distinct biological processes involved in phosphate homeostasis with the exploration of Klotho-FGF23-FGFR1 complex [161,162,163].

In conclusion, this review summarizes significant available information on vitamin D and FGF23 metabolism in the pediatric population, in normal conditions as well as in several diseases such as rickets, CKD and hypophosphatemic disorders. The article describes important published data, shows that the clinical interpretation of these findings is often unclear, and underlines the need for further studies aimed at disclosing the interrelation of vitamin D, Klotho and FGF23 in infants, children and adolescents.

## Figures and Tables

**Figure 1 ijms-24-06661-f001:**
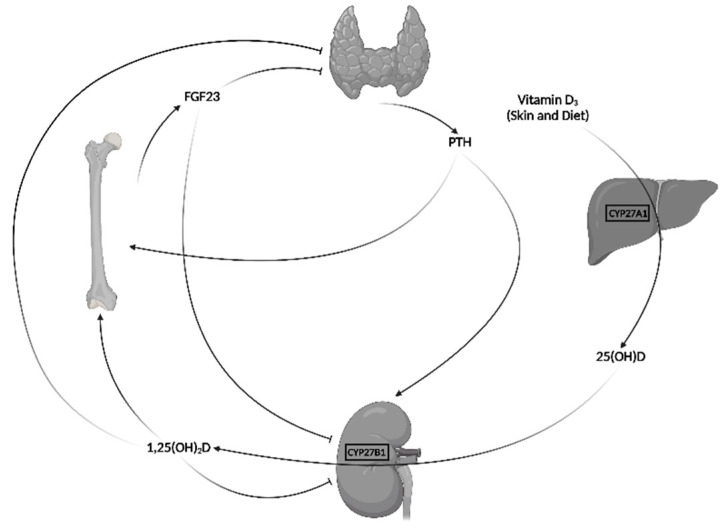
Parathyroid hormone (PTH)—vitamin D, fibroblast growth factor 23 (FGF23) feedback system. Lines ending in ͱndicate inhibitory effect. 25(OH)D: 25-hydroxyvitamin D. 1,25(OH)_2_D: 1,25-dihydroxyvitamin D. CYP27A1: cytochrome P450 family 27 subfamily A member 1. CYP27B1: cytochrome P450 family 27 subfamily B member 1. Figure created with BioRender.

**Table 1 ijms-24-06661-t001:** Age-specific and sex-specific pediatric reference intervals for calcium, phosphate, 1,25-dihydroxyvitamin D (1,25(OH)_2_D), parathyroid hormone (PTH), COOH-terminal (cFGF23) and intact fibroblast growth factor 23 (iFGF23) and Klotho [13,14,15,16,17,18,19,20,21,22].

Analyte	Units (SI)	First Author [Reference]	Gender	Age	No. of Samples	Reference Values		Lab Instrument
						Lower Limit (95% Confidence Interval)	Upper Limit (95% Confidence Interval)	
Calcium	mmol/L	Colantonio DA [13]	Females & males	0 to <1 Year	259	2.13 (2.10–2.17)	2.74 (2.70–2.78)	Abbott ARCHITECT c8000
				1 to <19 Years	897	2.29 (2.28–2.30)	2.63 (2.62–2.64)	
		Tahmasebi H [14]		0 to <2 Years	44	2.38 (2.31–2.44)	2.87 (2.82–2.92)	Siemens ADVIA XPT/1800
				2 to <5 Years	65	2.37 (2.35–2.40)	2.69 (2.67–2.72)	
				5 to <19 Years	409	2.28 (2.28–2.30)	2.55 (2.53–2.55)	
		Tahmasebi H [14]		0 to <2 Years	42	2.18 (2.09–2.24)	2.63 (2.59–2.67)	Siemens Dimension EXL
				2 to <19 Years	494	2.13 (2.13–2.15)	2.43 (2.40–2.43)	
Phosphate	mmol/L	Colantonio DA [13]	Females	0 to <15 Days	204	1.80 (1.73–1.90)	3.40 (3.29–3.47)	Abbott ARCHITECT c8000
				15 Days to <1 Year	144	1.54 (1.35–1.63)	2.72 (2.62–2.79)	
				1 to <5 Years	184	1.38 (1.29–1.45)	2.19 (2.11–2.39)	
				5 to <13 Years	352	1.33 (1.31–1.35)	1.92 (1.89–1.95)	
				13 to <16 Years	95	1.02 (0.98–1.06)	1.79 (1.73–1.84)	
				16 to <19 Years	187	0.95 (0.87–1.01)	1.62 (1.58–1.82)	
			Males	0 to <15 Days	204	1.80 (1.73–1.90)	3.40 (3.29–3.47)	
				15 Days to <1 Year	144	1.54 (1.35–1.63)	2.72 (2.62–2.79)	
				1 to <5 Years	184	1.38 (1.29–1.45)	2.19 (2.11–2.39)	
				5 to <13 Years	352	1.33 (1.31–1.35)	1.92 (1.89–1.95)	
				13 to <16 Years	95	**1.14 (1.10–1.17)**	**1.99 (1.93–2.04)**	
				16 to <19 Years	187	0.95 (0.87–1.01)	1.62 (1.58–1.82)	
Phosphate	mmol/L	Tahmasebi H [14]	Females	0 to <1 Year	125	1.36 (1.29–1.49)	2.49 (2.23–2.52)	Siemens ADVIA XPT/1800
				1 to <5 Years	86	1.42 (1.38–1.47)	1.99 (1.95–2.03)	
				5 to <13 Years	233	1.29 (1.23–1.32)	1.84 (1.81–1.94)	
				13 to <16 Years	56	1.05 (1.00–1.10)	1.68 (1.62–1.75)	
				16 to <19 Years	118	0.87 (0.82–0.92)	1.57 (1.53–1.61)	
			Males	0 to <1 Year	125	1.36 (1.29–1.49)	2.49 (2.23–2.52)	
				1 to <5 Years	86	1.42 (1.38–1.47)	1.99 (1.95–2.03)	
				5 to <13 Years	233	1.29 (1.23–1.32)	1.84 (1.81–1.94)	
				13 to <16 Years	56	**1.05 (0.95–1.13)**	**1.82 (1.77–1.89)**	
				16 to <19 Years	118	0.87 (0.82–0.92)	1.57 (1.53–1.61)	
		Tahmasebi H [14]	Females	0 to <1 Year	135	1.39 (1.39–1.55)	2.36 (2.23–2.36)	Siemens Dimension EXL
				1 to <5 Years	82	1.57 (1.53–1.60)	2.09 (2.05–2.14)	
				5 to <13 Years	202	1.49 (1.49–1.49)	2.00 (1.91–2.07)	
				13 to <16 Years	56	1.14 (1.10–1.18)	1.77 (1.69–1.84)	
				16 to <19 Years	116	0.93 (0.88–0.97)	1.64 (1.59–1.68)	
			Males	0 to <1 Year	135	1.39 (1.39–1.55)	2.36 (2.23–2.36)	
				1 to <5 Years	82	1.57 (1.53–1.60)	2.09 (2.05–2.14)	
				5 to <13 Years	202	1.49 (1.49–1.49)	2.00 (1.91–2.07)	
				13 to <16 Years	56	**1.08 (0.94–1.19)**	**1.86 (1.80–1.92)**	
				16 to <19 Years	116	0.93 (0.88–0.97)	1.64 (1.59–1.68)	
1,25(OH)_2_D	pmol/L	Higgins V [15]	Females & males	0 to <1 Year	105 83 185	77 (61–91)	471 (402–464)	DiaSorin LIAISON XL
				1 to <3 Years		113 (109–117)	363 (331–398)	
				3 to <19 Years		108 (104–110)	246 (225–355)	
PTH	pmol/L	Bailey D [16]	Females & males	6 Days to <1 Year	172	0.68 (0.42–0.91)	9.39 (7.93–15.48)	Abbott ARCHITECT i2000
				1 to <9 Years	221	1.72 (1.41–1.83)	6.68 (6.29–7.72)	
				9 to <17 Years	534	2.32 (2.18–2.40)	9.28 (8.52–9.82)	
				17 to <19 Years	104	1.7 (1.59–1.84)	6.4 (6.15–6.75)	
		Karbasy K [17]		0 to <1 Year	55	0.77 (0.58–0.98)	6.14 (5.57–6.86)	Beckman Dxl 800
				1 to <8 Years	194	1.25 (1.05–1.30)	5.80 (5.35–7.11)	
				8 to <19 Years	306	1.28 (1.06–1.39)	7.53 (7.13–8.56)	
		Higgins V [18]	Females	0 to <2 Weeks	45	0.57 (0.47–0.79)	11.50 (9.30–13.2)	Ortho Vitros 5600
				2 Weeks to <9 Years	203	1.23 (1.10–1.43)	7.14 (6.44–7.96)	
				9 to <15 Years	269	1.90 (1.72–1.94)	12.90 (10.10–13.90)	
				15 to <19 Years	84	1.44 (1.30–1.67)	6.67 (5.64–7.49)	
			Males	0 to <2 Weeks	45	0.57 (0.47–0.79)	11.50 (9.30–13.2)	
				2 Weeks to <9 Years	203	1.23 (1.10–1.43)	7.14 (6.44–7.96)	
				9 to <15 Years	269	1.90 (1.72–1.94)	12.90 (10.10–13.90)	
				15 to <19 Years	84	**1.66 (1.52–1.77)**	**8.41 (7.60–9.18)**	
		Bohn MK [19]	Females & males	0 to <1Month	50	0.7 (0.5–0.9)	6.3 (5.5–7.1)	Roche cobas 8000 e602
				1 to <12 Months	91	0.9 (0.8–1.1)	6.5 (5.7–7.3)	
				1 to <11 Years	199	1.2 (1.2–1.4)	6.3 (5.3–7.4)	
				11 to <19 Years	299	1.6 (1.3–1.7)	7.2 (6.7–8.8)	
cFGF23	RU/mL	Gkentzi D [20]	Females	8.4 (3.2–16.7) Years ^+^	77	53.36 ± 12.19 *		ELISA assay (Immutopics International)
			Males	8.3 (2.5–18) Years ^+^	82	49 ± 13.06 *		
iFGF23	pg/mL	Koyama S [21]	Females	12–13 Years	106	45.1 ± 21.2 *		ELISA assay
			Males		72	38.9 ± 31.6 *		
		Gkentzi D [20]	Females	8.4 (3.2–16.7) Years ^+^	77	36.8 (8.8–120) ^+^		ELISA assay (Kainos Laboratories)
			Males	8.3 (2.5–18) Years ^+^	82	33.15 (12.7–98.1) ^+^		
		Brescia V [22]	Females & Males	10 (1–18) Years ^+^	115	61.21 (58.63–63.71) **		DiaSorin LIAISON XL
Klotho	pg/mL	Gkentzi D [20]	Females	8.4 (3.2–16.7) Years ^+^	77	2487 (964–5866) ^+^		ELISA assay (IBL America)
			Males	8.3 (2.5–18) Years ^+^	82	1692 (372–5694) ^+^		

SI: International system of units. No.: number. Male-specific reference intervals are highlighted in bold. ^+^: median with lower and upper value. *: mean ± standard deviation (SD). **: upper reference limit (90% confidence interval).

**Table 2 ijms-24-06661-t002:** Publications showing early manifestations induced by fibroblast growth factor 23 (FGF23) deficiency or excess in clinical case reports or animal models [26,30,31,32,33,34].

Case Reports in Humans			
Disease/Cause	Presentation	Age at Diagnosis	First Author & Year
Hereditary hyperphosphatemic calcinosis (FGF23 deficiency)	Calcified mass	18 days	Slavin RE, 2012 [30] Polykandriotis EP, 2004 [31]
X-linked hypophosphatemic rickets caused by mutations in the *PHEX* gene (excess of FGF23)	- Hypophosphatemia - Hyperphosphaturia	9 days	Moncrieff MW, 1982 [32]
**Animal Models**			
**Cause**	**Presentation**	**Age at Diagnosis**	**First Author & Year**
Mice lacking FGF23 or its co-receptor Klotho	- Hyperphosphatemia - Reduced renal phosphorus excretion - High calcitriol levels - Marked hypercalcemia - Soft tissue calcifications - Skeletal deformities - Death (10–12 days of life)	4–5 days	- Ma Y, 2017 [26] - Sitara D, 2004 [33]
*Phex*-null mice (Excess of FGF23)	- Hypophosphatemia - Increased renal phosphorus excretion - Short and abnormal bones	12 h	- Ma Y, 2017 [26] - Ma Y, 2014 [34]

**Table 3 ijms-24-06661-t003:** Skeletal signs of nutritional rickets: denomination and illustrative drawings.

Craniotabes (soft skull bones)	** 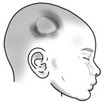 **
Delayed closure of the fontanelles Parietal and frontal bossing. Caput cuadratum	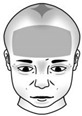
Enlargement of the costochondral junction: “rachitic rosary”	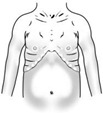
Softened lower ribs, tractioned by the diaphragmatic attachments: Harrison’s sulcus	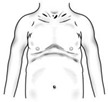
Widening of the wrist and bowing of the distal radius and ulna	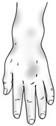
Incurvations of long bones: varus, valgus genum and windswept deformity	
Kyphosis	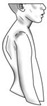
Pelvic deformities: narrowing	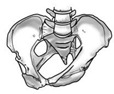

**Table 4 ijms-24-06661-t004:** Oral doses of vitamin D for treatment of nutritional rickets.

Age	Dose (IU) */Day for 3 Months	Single Dose (IU)	Maintenance Dose/Day (IU)
<3 months	2000	Not available	400
3 months–1 year	2000	50,000	400
1–12 years	3000–6000	150,000	600
>12 years	6000	300,000	600

* IU: International Units (1 μg = 40 IU).

**Table 5 ijms-24-06661-t005:** Most significant randomized controlled trials (RCT) and systematic reviews published in the last 5 years analyzing the effect of vitamin D (VD) supplementation in patients with systemic diseases [52,53,54,55,56,57,58,59,60,61,62,63,64,65,66].

Publication 1st Author, Year [Reference]	Number of Studies and of Included Individuals	Age of Participants (Years)	Disease and Analyzed Variable	Findings and Conclusions
Barbarawi,2019 [52]	21 83,291	Adults	Cardiovascular disease (CVD) events and all-cause mortality 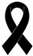	VD supplementation did not reduce major CVD events, individual CVD endpoints (myocardial infarction, stroke, CVD mortality), or all-cause mortality. VD supplementation did not confer cardiovascular protection and is not indicated for this purpose.
Manson, 2019 [53]	1 25,871	≥50	Cancer and CVD 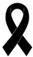	VD supplementation did not result in a lower incidence of invasive cancer or CVD events than placebo.
Zhang, 2019 [54]	50 74,655	Adults	Global mortality and cancer death 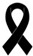	VD supplementation alone was not associated with decrease in all cause mortality compared with placebo or no treatment. VD supplementation reduced the risk of cancer death by 15%.
Zheng, 2019 [55]	5 490	Adults	Systemic lupus erythematosus (SLE) 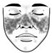	VD supplementation increased serum 25OHD levels, improved fatigue, and was well-tolerated. However, it did not have significant effects in decreasing the positivity of anti-dsDNA and disease activity.
Pittas, 2019 [56]	1 2423	≥30	Type 2 diabetes 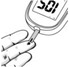	Among persons at high risk for type 2 diabetes not selected for VD insufficiency, VD supplementation at a dose of 4000 IU per day did not result in significantly lower risk of diabetes than placebo.
Arihiro, 2019 [57]	1 223	18–80	Influenza and upper respiratory infections in patients with inflammatory bowel disease 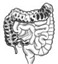	VD supplementation may have a preventive effect against upper respiratory infection in patients with inflammatory bowel disease, but may worsen the symptoms of ulcerative colitis.
De Boer, 2019 [58]	1 1312	Adults	Kidney function in type 2 diabetes. 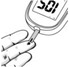	The findings do not support the use of VD or omega-3 fatty acid supplementation for preserving kidney function in patients with type 2 diabetes.
Ganmaa, 2020 [59]	1 8851	6–13	Tuberculosis (TB) 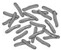	VD supplementation did not result in a lower risk of TB infection, TB disease, or ARI than placebo among VD–deficient school children.
Rawat, 2021 [60]	3 467	Adults	COVID-19 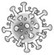	VD supplementation did not reduce major health related outcomes like mortality, ICU admission rates and mechanical ventilation.
Bassatne, 2021 [61]	3356	Adults	COVID-19 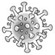	The evidence available to-date is insufficient to make any recommendations for high doses of VD to either prevent or treat COVID-19 complications.
Jolliffe, 2021 [62]	4675,541	0–95	Acute respiratory infections (ARI) 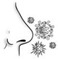	VD supplementation was safe and overall reduced the risk of ARI compared with placebo, although the risk reduction was small.
Theodoridis, 2021 [63]	4252	18–84	Psoriasis 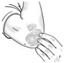	A favorable effect of oral VD supplementation in patients with psoriasis could not be verified.
Juhász, 2021 [64]	8301	Children and adults	Cystic fibrosis 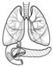	The intervention group had significantly higher serum 25OHD levels, but there were no significant differences found in the quantitative synthesis of clinical outcomes.
Hahn, 2022 [65]	125,871	Adults	Autoimmune disease (AD) risk	VD supplementation for 5 years, with or without omega 3 fatty acids, reduced AD by 22%, while omega 3 fatty acid supplementation with or without VD reduced the AD rate by 15% (not statistically significant). Both treatment arms showed better effects than placebo.
Luo, 2022 [66]	71948	Adults and children ≤ 5	Allergic diseases 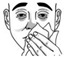	VD supplementation in pregnant women or children from birth to 5 years of age did not have any effect on the primary prevention of allergic diseases.

CVD: cardiovascular disease. SLE: systemic lupus erythematosus. TB: tuberculosis. ARI: acute respiratory infections. AD: autoimmune disease.

**Table 6 ijms-24-06661-t006:** Descriptive characteristics of elevated fibroblast growth factor 23 (FGF23) levels found in pediatric patients with chronic kidney disease (CKD) [83,84,85,86,87,88,89,90,91,92,93,94].

High Serum FGF23	References
Intact and C-terminal FGF23 are elevated	[84]
Wide range, no daytime variability	[83]
Found in all stages of CKD	[83,85]
Higher concentrations in advanced CKD stages	[86,87,88,89]
Chronologically precedes hyperphosphatemia and hyperparathyroidism	[88]
Occurs after reduction of serum klotho levels	[87]
Correlation with other variables in serum or plasma: directly with phosphorus, PTH and calcium—phosphorus product; inversely with 1,25(OH)_2_D. No consistent correlation with calcium or 25OHD.	[90,91,92,93,94]

PTH: parathyroid hormone. 1,25(OH)_2_D: 1,25-dihydroxyvitamin D. 25OHD: 25-hydroxyvitamin D.

**Table 7 ijms-24-06661-t007:** Characteristics of inherited and acquired types of rickets. Adapted from Bitzan et al. [114], Haffner et al. [115], and Tiosano et al. [116].

Disorder (OMIM#)	Gene	Ca	Pi	ALP	U_Ca/Cr_	U_P/Cr_	FGF23	PTH	25OHD	1,25 (OH)_2_D	Pathogenesis
** *Rickets with High PTH Levels* **											
Nutritional rickets	NA	N, ↓	N, ↓	↑↑↑	↓	Varies	N, ↓	↑↑↑	↓↓, N	Varies	Vitamin D deficiency
Vitamin-D-dependent rickets type 1A (OMIM#264700)	*CYP27B1*	↓	N, ↓	↑↑↑	↓	Varies	N, ↓	↑↑↑	N	↓	Impaired synthesis of 1,25(OH)_2_D
Vitamin-D-dependent rickets type 1B (OMIM#600081)	*CYP2R1*	↓	N, ↓	↑↑↑	↓	Varies	N, ↓	↑↑↑	↓↓	Varies	Impaired synthesis of 25OHD
Vitamin-D-dependent rickets type 2A (OMIM#277440)	*VDR*	↓	N, ↓	↑↑↑	↓	Varies	N, ↓	↑↑↑	N	↑↑	Impaired signalling of the VDR
Vitamin-D-dependent rickets type 2B (OMIM#600785)	*HNRNPC*	↓	N, ↓	↑↑↑	↓	Varies	N, ↓	↑↑↑	N	↑↑	Impaired signalling of the VDR
Vitamin-D-dependent rickets type 3 (OMIM#619073)	*CYP3A4*	↓	N, ↓	↑↑↑	↓	Varies	?	↑↑↑	↓	↓	↑ Inactivation of 1,25(OH)_2_D
** *Rickets due to phosphate deficiency* **											
Dietary deficiency or impaired bioavailability	NA	N, ↑	↓	↑, ↑↑	?	↓	N, ↓	N	N	N, ↑	Phosphate deficiency
** *Rickets with renal tubular phosphate wasting due to elevated FGF23 levels and/or signalling* **											
X-linked hypophosphatemia (OMIM#307800)	*PHEX*	N	↓	↑, ↑↑	↓	↑	↑, N	N, ↑	N	N, ↓	↑ FGF23 expression in bone
Autosomal dominant hypophosphatemic rickets (OMIM#193100)	*FGF23*	N	↓	↑, ↑↑	↓	↑	↑, N	N, ↑	N	N, ↓	FGF23 protein resistant to degradation
Autosomal recessive hypophosphatemic rickets 1 (OMIM#241520)	*DMP1*	N	↓	↑, ↑↑	↓	↑	↑, N	N, ↑	N	N, ↓	↑ FGF23 expression in bone
Autosomal recessive hypophosphatemic rickets 2 (OMIM#613312)	*ENPP1*	N	↓	↑, ↑↑	↓	↑	↑, N	N, ↑	N	N, ↓	↑ FGF23 expression in bone
Raine syndrome associated (OMIM#259775)	*FAM20C*	N	↓	↑, ↑↑	?	↑	↑, N	N, ↑	N	N, ↓	↑ FGF23 expression in bone
McCune-Albright syndrome (OMIM#174800)	*GNAS*	N, ↓	↓	↑, ↑↑	↓	↑	N, ↑	N, ↑	N	N, ↓	↑ FGF23 expression in bone
Tumor-induced osteomalacia	NA	N, ↓	↓	↑, ↑↑	↓	↑	N, ↑	N, ↑	N	N, ↓	↑ FGF23 expression in tumoral cells
Cutaneous skeletal hypophosphatemia syndrome (OMIM#163200)	*RAS*	N, ↓	↓	↑, ↑↑	↓	↑	N, ↑	N, ↑	N	N, ↓	Unknown
Osteoglophonic dysplasia (OMIM#166250)	*FGFR1*	N	↓	↑, N	N	↑	N	N, ↑	N	N, ↓	↑ FGF23 expression in bone
Hypophosphatemic rickets and hyperparathyroidism (OMIM#612089)	*KLOTHO*	↑	↓	↑, ↑↑	N	↑	↑	↑↑	N	N, ↓	Unknown
** *Rickets due to primary renal tubular phosphate wasting* **											
Hereditary hypophosphatemic rickets with hypercalciuria (OMIM#241530)	*SLC34A3*	N, ↑	↓	↑, ↑↑	N, ↑	↑	↓	N, ↓	N	N, ↑	Loss of function of NaPi2c in the proximal tubule
X-linked recessive hypophosphatemic rickets (OMIM#300554)	*CLCN5*	N	↓	↑, ↑↑	N, ↑	↑	Varies	Varies	N	Varies	Loss of function of CLCN5 in the proximal tubule
Hypophosphatemia and nephrocalcinosis (OMIM#612286), Fanconi reno-tubular syndrome 2 (OMIM#613388) and Hypercalcemia infantile 2 (OMIM# #616963)	*SLC34A1*	N, ↑	↓	↑, ↑↑	N, ↑	↑	↓	N, ↓	N	N, ↑	Loss of function of NaPi2a in the proximal tubule
Cystinosis (OMIM#219800) and other hereditary forms of Fanconi syndrome	*CTNS*	N, ↓	↓	↑, ↑↑	N, ↑	↑	Varies	Varies	N	Varies	Cysteine accumulation in the proximal tubule
Iatrogenic proximal tubulopathy	NA	N, ↑	↓	↑, ↑↑	Varies	↑	↓	Varies	N	N, ↑	Drug toxicity

NA: not applicable; N: normal; ?: unknown; ↓: decreased; ↓↓: very decreased; ↑: elevated; ↑↑ or ↑↑↑: very elevated; Ca: serum levels of calcium; Pi: serum levels of phosphate; ALP: alkaline phosphatase; UCa/Cr: urinary calcium to creatinine ratio; UP/Cr: urinary phosphate to creatinine ratio; FGF23: fibroblast growth factor 23; PTH: parathyroid hormone; 1,25(OH)_2_D: 1,25-dihydroxyvitamin D; 25OHD: 25-hydroxyvitamin D.

**Table 8 ijms-24-06661-t008:** Most significant studies that analyze the effects of treatment and the interaction between fibroblast growth factor 23 (FGF23), klotho and vitamin D in patients with phosphate disorders [129,130,131,132,133,134,135,136,137,138,139,140,141,142,143,144,145,146,147,148,149,150,151,152,153,154,155,156,157,158,159,160].

Reference	Treatment	Age	Findings and Conclusions
X-linked hypophosphatemia			
Imel, 2010 [129] Carpenter, 2010 [130] Rodríguez-Rubio, 2021 [131]	Conventional therapy (phosphate and vitamin D derivatives)	2–41 y 9–60 y 3 mo–8 y	Traditional treatment did not correct hypophosphatemia and was associated with a risk of hyperparathyroidism. FGF23 was greater in XLH than in controls and greater in treated XLH subjects compared with XLH subjects not receiving phosphate and calcitriol. Slightly lower fasting values for serum phosphate and 1,25(OH)_2_D were found in treated subjects compared with untreated subjects. Moreover, a strong positive correlation between FGF23 and PTH was also found in XLH subjects, suggesting aberrant PTH secretion. Serum klotho declined with age and had circadian variation but was normal in XLH.
Carpenter, 2014 [132]	Paricalcitol	10–69 y	PTH decreased from baseline in subjects receiving paricalcitol and increased in subjects receiving placebo. FGF23 level also increased in paricalcitol-treated patients, despite which fasting serum phosphorus increased and phosphaturia decreased in the treatment group. 1,25(OH)_2_D did not change in subjects receiving paricalcitol.
Alon, 2008 [133]	Cinacalcet	6–19 y	Oral phosphate load increased serum phosphate and increased PTH. FGF23 significantly increased and 1,25(OH)_2_D decreased. The concomitant administration of phosphate and cinacalcet resulted in suppression of PTH, greater increase in serum phosphate and decrease in phosphaturia, presumably because of the greater suppression of PTH. Phosphaturia did not change in patients treated with combined phosphate and calcitriol.
Carpenter 2018 [134] Whyte 2019 [135] Imel 2019 [136] Martin Ramos 2020 [137] Linglart 2022 [138]	Burosumab	5–12 y 1–4 y 1–12 y 6–16 y 5–12 y	Burosumab increased the serum levels of inorganic phosphate and 1,25(OH)_2_D and reduced phosphaturia in the long-term, despite prior treatment with phosphate salts and activated forms of vitamin D. Moreover, burosumab was more effective than continuing conventional therapy.
Lecoq, 2020 [139]	No treatment	Adults	Patients have increased PTH compared with healthy controls matched for sex, age, and vitamin D status, suggesting hyperparathyroidism in XLH is associated with disruption of the physiological regulation of PTH secretion, although no correlation between PTH and FGF23 was shown in this study.
Insogna, 2018 [140]	Burosumab	Adults	Plasma PTH decreased in the burosumab group and increased in the placebo group. Administration of burosumab also increased serum 1,25(OH)_2_D.
**Autosomal dominant hypophosphatemic rickets**			
Imel, 2007 [141]	Conventional therapy	17–83 y	Elevated FGF23 concentrations were associated with hypophosphatemia in the patients, and resolution of the phenotype was associated with normalization of FGF23.
Imel, 2011 [142]	No treatment	14–85 y	Serum phosphate and 1,25(OH)_2_D correlated negatively with C-terminal FGF23 and intact FGF23 in patients. Low serum iron is associated with elevated FGF23 in ADHR. However, in controls, low serum iron was also associated with elevated C-terminal FGF23, but not intact FGF23, suggesting cleavage maintains homeostasis despite increased FGF23 expression.
Imel, 2020 [143]	Oral iron supplements. Conventional therapy was allowed	Adults	Oral iron administration normalized FGF23 and phosphorus in iron-deficient ADHR subjects.
**Other disorders with elevated FGF23**			
Bai, 2022 [144]	Burosumab	Adults	In adults with ARHR1, burosumab resulted in normalization of serum phosphate and 1,25(OH)_2_D. PTH levels remained notably stable after an initial rise.
Yamamoto, 2005 [145] Florenzano, 2019 [146]	Bisphosphonates	2–80 y	Plasma FGF23 levels were significantly increased in patients with MAS compared to normal controls. Plasma FGF23 levels showed significant negative correlation with serum phosphate concentrations. Bisphosphonate treatment did not significantly impact the age-dependent decrease in bone turnover markers, including FGF23.
Gladding, 2021 [147] Apperley, 2022 [148]	Burosumab	5–8 y	Burosumab achieved sustained normalization of serum phosphorus and reduced PTH levels in patients with MAS.
Khadora, 2021 [149] Merz, 2022 [150]	Burosumab	3 y	In patients with CSHS, Burosumab led to normalization of serum phosphate and slight decline of PTH levels, without changes in 1,25(OH)_2_D.
Imanishi, 2021 [151] Jan de Beur, 2021 [152]	Burosumab	Adults	In adults with TIO, burosumab was associated with increased serum levels of phosphate and 1,25(OH)_2_D, as well as reduced phosphaturia.
**Disorders with primary renal tubular phosphate wasting**			
Kremke, 2009 [153] Yu 2012 [154] Schlingmann, 2016 [155] Gordon, 2020 [156] Gurevich, 2021 [157] Christensen, 2021 [158] Molin, 2021 [159] Magen, 2010 [160]	Phosphate supplements	8–25 y 2–46 y 1 mo–1.5 y 14–68 1 mo–1 y8–13 y 1 day–81 y Adults	Overlapping phenotypes associated with *SLC34A1*, *SLC34A3* and *CYP24A1* mutations have been described, and that not all the patients showed improvements in hypercalciuria and nephrocalcinosis, despite improvement in hypercalcemia and 1,25(OH)_2_D levels. Most of these infants presented with severe hypercalcemia, profound renal phosphate-wasting and suppressed serum PTH, as well elevated serum 1,25(OH)_2_D during hipercalcemia. An attenuation of renal phosphate wasting with advancing age has been observed and vitamin D deficiency can mask some of the characteristic laboratory findings. Treatment with oral phosphate supplements restored serum levels of phosphate and FGF23 enabling a normalization of 1,25(OH)_2_D.

ADHR: autosomal dominant hypophosphatemic rickets; ARHR1: autosomal recessive hypophosphatemic rickets 1; CSHS: cutaneous skeletal hypophosphatemia syndrome; FGF23: fibroblast growth factor 23; HHRH: Hereditary hypophosphatemic rickets with hypercalciuria; iFGF23: intact fibroblast growth factor 23; 1,25(OH)_2_D: 1,25-dihydroxyvitamin D; MAS: McCune-Albright syndrome; Mo: months; PTH: parathyroid hormone; TIO: Tumor-induced osteomalacia; XLH: X-linked hypophosphatemia; Y: years.

## Data Availability

Not applicable.

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
