# Peer review of "Mineral Metabolism in Children: Interrelation between Vitamin D and FGF23"

_ijms, 2023, doi:10.3390/ijms24076661_

Round 1

Reviewer 1 Report

A relevant study, named “Mineral metabolism in children: interrelation between vitamin D and FGF23” that revised mineral homeostasis in the pediatric age.

This review has some important points, emphasizing the Vitamin D deficiency, chronic kidney disease and hypophosphatemic disorders in the pediatric age. However, in my view, certain points need to be considered. Clarify that this is a narrative review in the abstract, expand the discussion of reference values in the pediatric age for the mineral metabolism, and display all acronyms and images in the subtitles.

Author Response

First of all thank you for your review. I have activated change control to make it easier to review and I have implemented your suggestions:

- Clarified that this is a narrative review in abstract.

- Added paragraphs on paediatric reference values for mineral metabolism.

- Added all acronyms at the bottom of tables and images.

In addition, I have rewritten a few paragraphs and two additional references due to the suggestions of reviewer 2.

Reviewer 2 Report

This manuscript reviews a large body of published information and is a comprehensive collection of current knowledge relating the endocrine systems of vitamin D and of FGF23 in growing children. There are a couple of small points that might be addressed by the authors:

1.      Lines 107-108: “…stimulating the enzyme 24-hydroxylase, that degrades 1,25(OH)2D.” It is commonly stated that 24-hydroxylation of both 25(OH)D and 1,25(OH)2D are processes in the inactivation and eventual degradation of the vitamin D structure. However, there are now quite a number of papers that provide evidence that 24,25(OH)2D is a renal metabolite of 25(OH)D that has a specific function in the mineralisation of bone during growth and in the repair of bone fractures. It would be appropriate to mention such reports in a review of this type. One paper that could lead into this literature is by Seo et al. (1997) Endocrinology 138(9), 3864-3872.

2.      Lines 58-60: The cells and tissues that produce FGF23 are listed and at line 67 the role of the α-Klotho co-receptor in increasing the affinity of FGF23 for its receptor is mentioned. There does not appear to be any description of Klotho in the text – its site of production and its overall functions. There is some discussion of Klotho in lines 380-390, but it would helpful in a review of this type to provide some systematic information about the physiology of Klotho, particularly as its function seems to be closely linked with that of FGF23.

Author Response

First of all thank you for your review. I have activated change control to make it easier to review and I have implemented your suggestions:

- I have added a paragraph on 24,25(OH)2D and suggested reference

- I have added a paragraph on klotho

In addition, I have rewritten a few paragraphs and I have added some suggestions from the reviewer 1.